# Feasibility of Simultaneous Multislice Acceleration Technique in Readout-Segmented Echo-Planar Diffusion-Weighted Imaging for Assessing Rectal Cancer

**DOI:** 10.3390/diagnostics13030474

**Published:** 2023-01-28

**Authors:** Mi Zhou, Hong Pu, Mei-Ning Chen, Yu-Ting Wang

**Affiliations:** 1Department of Radiology, Sichuan Provincial People’s Hospital, University of Electronic Science and Technology of China, Chengdu 610072, China; 2Department of MR Scientific Marketing, Siemens Healthineers, Shanghai 200135, China

**Keywords:** rectal cancer, simultaneous multislice, apparent diffusion coefficient, readout-segmented echo-planar imaging

## Abstract

Background: Readout-segmented echo-planar imaging (rs-EPI) with simultaneous multislice (SMS) technology has been successfully applied to tumor research in many organs, but no feasibility study in rectal cancer has been reported, and the optimal acceleration of SMS with rs-EPI in rectal cancer has not been well determined yet. Objective: To investigate the feasibility of SMS rs-EPI of rectal cancer with different acceleration factors (AF_s_) and its influence on image quality, acquisition time and apparent diffusion coefficients (ADC_s_) in comparison to conventional sequences. Methods: All patients underwent rs-EPI and SMS rs-EPI with AF_s_ of 2 and 3 (2 × SMS rs-EPI and 3 × SMS rs-EPI, respectively) using a 3T scanner. Acquisition times of the three rs-EPI sequences were measured. Image qualitative parameters (5-point Likert scale), signal-to-noise ratio (SNR), contrast-to-noise ratio (CNR), geometric distortion, and apparent diffusion coefficient (ADC) values of the three sequences were compared. Results: A total of eighty-three patients were enrolled in our study. rs-EPI and 2 × SMS rs-EPI offered equivalently high overall image quality with a scan time reduction to nearly half (rs-EPI: 137 s, 2 × SM rs-EPI: 60 s). 3 × SMS rs-EPI showed significantly poorer image quality (*p* < 0.05). ADC values were significantly lower in 3 × SMS rs-EPI compared to rs-EPI in rectal tumors and normal tissue (tumor tissue: rs-EPI 1.19 ± 0.21 × 10^−3^ mm^2^/s, 3 × SMS rs-EPI 1.10 ± 0.26 × 10^−3^ mm^2^/s, *p* < 0.001; normal tissue: rs-EPI 1.68 ± 0.13 × 10^−3^ mm^2^/s, 3 × SMS rs-EPI 1.54 ± 0.20 × 10^−3^ mm^2^/s, *p* < 0.001). Conclusions: SMS rs-EPI using an AF of 2 is feasible for rectal MRI resulting in substantial reductions in acquisition time while maintaining diagnostic image quality and similar ADC values to those of rs-EPI when the slice distance and number of shots are the same among three rs-EPI sequences.

## 1. Introduction

Colorectal cancer is a leading cause of high morbidity and mortality worldwide, where 30–35% of colorectal cancers occur in the rectum [1,2]. Magnetic Resonance Imaging (MRI) is the preferred diagnostic tool for the local staging of rectal cancer, which provides excellent soft-tissue contrast and may be used to assess treatment response and detect postoperative local recurrence [3]. Diffusion Weighted Imaging (DWI) is a fast, non-enhanced MRI technique that does not require a contrast agent for assessing the diffusive movement of water molecules in tissues. Employing DWI with conventional MRI improves diagnostic confidence and may be used to predict rectal cancer [4,5]. Additionally, the Apparent Diffusion Coefficient (ADC) from traditional DWI provides biological implications for cell density and can identify tumor progression [6].

Due to its high speed, single-shot echo-planar imaging (ss-EPI) DWI is widely used in radiology practice, with K-space filling after a single excitation. However, due to the slow filling along the phase-encoding direction, the low bandwidth in this direction results in blurred and distorted images. Additionally, ss-EPI is highly susceptible to magnetic field inhomogeneities and may result in geometric distortions [7]. Parallel acquisition techniques and multi-shot EPI sequences, such as readout-segmented echo-planar imaging (rs-EPI), can reduce this limitation [8]. rs-EPI divides K-space into segments along the readout direction to shorten the echo interval [8]. Previous studies have also shown that rs-EPI is superior to ss-EPI in assessing lesions in patients with rectal cancer and improving image quality; lesion conspicuity, overall image quality, geometric distortion, and distinction of anatomic structures were significantly better in rs-EPI [9].

rs-EPI, however, is limited by its long scan time, which can cause motion artifacts due to bowel peristalsis and patient discomfort. Based on the principle of simultaneously exciting multiple slices, a new method using simultaneous multislice (SMS) technology has been proposed to reduce DWI scanning time [10]. This technique involves simultaneous excitation and acquisition of multiple slices, enabling a reduction in scan time based on a greater number of simultaneously excited slices as determined by the acceleration factor (AF) [11]. In this technique, the number of slices acquired over the same pulse repetition time can be increased by setting a higher AF without the need for an increase in the gradient demand [12]. The SMS combined with DWI has been successfully applied to the kidney, liver, pancreas, breast, and other organs [13,14,15] with promising results. According to Park et al. [11], SMS combined with ss-EPI with an AF of 2 yields an image quality and stable ADC values comparative to those of conventional ss-EPI. In theory, image acquisition in sms-DWI can be accelerated to a large extent by increasing the applied AFs. However, an arbitrary increase of the AF is expected to come along with a deterioration of image quality [11,13]. To ascertain the optimal ratio between scan time minimization and diagnostic image quality, the use of higher AFs and the resulting image quality need to be investigated systematically. What is more, there are no studies using SMS in combination with rs-EPI in rectal cancer as far as we know, and the impact of rs-EPI combined with different SMS AF_s_ on the image quality and ADC values of rectal cancer is also unknown.

Thus, the purpose of this study was to investigate the feasibility of SMS rs-EPI of rectal cancer with different acceleration factors (AF_s_) and its influence on image quality, acquisition time and apparent diffusion coefficients (ADC_s_) in comparison with conventional sequences.

## 2. Materials and Methods

### 2.1. Patients

This retrospective study was approved by the Institutional Review Board. Informed consent was obtained from all patients.

Based on a clinical history and physical examination results, 203 patients with clinically suspected RC were enrolled between December 2021 and June 2022. Patients with non-mucinous rectum adenocarcinoma confirmed by biopsy after endoscopically guided biopsy were included. The exclusion criteria were: (1) patients diagnosed with RC after undergoing preoperative radiotherapy and chemotherapy (n = 53); (2) tumor not visible on the MRI image (n = 14); (3) serious image artifacts, ROI measurement cannot be performed well (n = 5 for rs-EPI, n = 9 for 2 × SMS rs-EPI, n = 17 for 3 × SMS rs-EPI); (4) surgical contraindications (n = 5); (5) unresectable or metastatic diseases (n = 8); and (6) the existence of mucinous cystadenoma, whose cell density is rather low and therefore exhibits high ADC values (n = 9).

### 2.2. MRI Imaging

A 3.0T MRI scanner (MAGNETOM Vida, Siemens Healthineers, Shanghai, China) with an 18-channel body array coil was used. Bowel cleansing with enema was performed 50 min before the examination. Patients were then administered 20 mg of scopolamine butylbromide (Buscopan, Boehringer Ingelheim, Shanghai, China) intramuscularly 30 min before testing to minimize bowel motion. Three scan sequences were performed for each subject: rs-EPI, SMS rs-EPI sequence with an acceleration factor of 2 (2 × SMS rs-EPI), and SMS rs-EPI (3 × SMS rs-EPI) with an acceleration factor of 3, all in free-breathing scanning mode, with the scanning direction perpendicular to the diseased intestine. Conventional MRI scan sequences were: (1) coronal breath-hold Truifi sequence (auxiliary positioning); (2) Transverse T1 VIBE DIXON sequence; and (3) Transverse high-resolution TSE T2WI sequence. The number of scanned slices was 24. The parameters of the three rs-EPI sequences are seen in Table 1. The fat press method was Spectral Attenuated Inversion Recovery (SPAIR). The b values of all DWI sequences were 0 and 1000 s/mm^2^ [9,16,17]. No intravenous contrast was used in this study.

### 2.3. Image Quality-Qualitative Analysis

All images were imported into the Syngo.via workstation, and the rs-EPI images with 2 b values were processed using the workstation’s software to obtain ADC maps. All images were transferred to the picture archiving and communication system (PACS). Image analysis was independently performed by two physicians (with 6 and 16 years of diagnostic experience in MRI of the rectum, respectively) who were blinded to pathological findings and MRI sequences. Subjective parameters of image quality (lesion conspicuity, overall image quality, and margin sharpness) were evaluated based on a 5-point Likert scale, with higher scores indicating better image quality. The results of the subjective scores were averaged from the scores of the two observers. The subjective parameters were assessed in b = 0 s/mm^2^, b = 1000 s/mm^2^, and ADC map of three rs-EPI sequences, respectively. Lesions were evaluated as follows: (1) the rectal cancer lesion was difficult to detect; (2) the lesion was minimally recognizable; (3) it was moderately visible; (4) it was well visible; (5) it showed excellent visibility. The overall image quality was evaluated as follows: (1) non-diagnostic; (2) poor; (3) satisfactory; (4) good; (5) excellent. The margin sharpness was evaluated as follows [18]: (1) not sharp; (2) slightly sharp; (3) moderately sharp; (4) good sharpness; (5) excellent sharpness [18].

### 2.4. Image Quality-Quantitative Analysis

For quantitative analysis, the two radiologists performed the region of interest (ROI) measurement in consensus. First, two ROIs were drawn on the DWI images with b = 0 s/mm^2^ to include the largest possible areas of rectal cancer and normal rectal wall of the three groups, respectively. Then, the ROIs were copied to the DWI images with b = 1000 s/mm^2^ for rs-EPI and SMS rs-EPI to obtain the mean signal intensity and standard deviation (Figure 1). Signal-to-noise ratio (SNR) and contrast-to-noise (CNR) were calculated according to the following equations [19]:$$\mathrm{SNR}=\;\frac{S_{lesion}}{SD_{background}}$$
$$\mathrm{CNR}=\;\frac{\left|S_{lesion}-S_{normal\;tissue}\right|}{\sqrt{SD_{lesion}^{}2+SD_{normal\;tissue}^{}2}}\;,$$
where $\;S_{lesion}$ represents the signal intensity of the rectal mass, $S_{normal\;tissue}$ represents signal intensity of the normal rectal wall, and $SD_{background}$, $SD_{lesion}^{},\;\;SD_{normal\;tissue}^{}$ represent the standard deviation of signal intensity of the background area, rectal mass, and normal rectal wall, respectively.

The above quantitative parameters were measured three times to obtain the average values of the two radiologists.

The above ROI was copied onto the ADC image to measure ADC values of rectal masses. In addition, the ROI also selected the non-mass region of the rectum, and the ADC values of normal tissue were measured. The thickest level of the intestinal wall in T2WI was selected for measurement. Geometric distortion was evaluated by comparing lesion lengths in the largest areas of rectal cancer between axial T_2_WI images and the b = 0 s/mm^2^ and b = 1000 s/mm^2^ of three rs-EPI sequences. Anterior–posterior (AP) length and left–right (LR) width of the lesion were measured, and the distortion rate was calculated (rs − EPI distance − T_2_WI distance)/T_2_WI distance × 100 [20].

### 2.5. Statistical Analysis

SPSS version 22 (IBM Corporation) was used for analysis. Measures are expressed as $\bar{X}\pm s$. There was inter-reader agreement of the quantitative parameters (ADC and the qualitative image scores given by the two radiologists were assessed by calculating their respective intra-class correlation coefficients (ICC) (0.00–0.20, poor agreement; 0.21–0.40, fair agreement; 0.41–0.60, moderate agreement; 0.61–0.80, good agreement; and 0.81–1.00, excellent agreement) [21]. The Kolmogorov–Smirnov test was used to assess the normality of the quantitative data distribution. Comparisons of image quality, SNR, CNR, and geometric distortion, and the ADC values of the three groups of rs-EPI sequences were performed using the Freidman test with the Bonferroni post hoc test [11,18,22]. A *p* value < 0.05 indicated a significant difference.

## 3. Results

Eighty-three patients (55 males and 28 females, 22–76 years old, with a mean age of 54.6±11.9 years old) who underwent MRI were included in this study. Results of the histochemical assay of rectal cancer were: 13 (15.7%) were well differentiated, 64 (77.1%) moderately differentiated, and 6 (7.2%) poorly differentiated. The clinicopathological features of the patients are shown in Table 2.

### 3.1. Acquisition Time

The scanning time of 2 × SMS rs-EPI was 60 s, which was 56.2% shorter than that of 137 s for rs-EPI sequences; the scanning time of 3 × SMS rs-EPI was 51 s, which was 72.8% shorter than that of rs-EPI sequences.

### 3.2. Qualitative Analysis

The inter-reader agreement was excellent for lesion conspicuity (ICC = 0.843 (0.763–0.896), 0.947 (0.919–0.965) and 0.772 (0.669–0.846)), overall image quality (ICC = 0.895 (0.842–0.931), 0.847 (0.773–0.898) and 0.844 (0.768–0.896)), margin sharpness (ICC = 0.929 (0.891–0.954), 0.935 (0.901–0.957) and 0.942 (0.912–0.962)) in rs-EPI, 2 × SMS rs-EPI and 3 × SMS rs-EPI, respectively.

The scores of lesion conspicuity, overall image quality and margin sharpness on DWI images and ADC map were significantly different among rs-EPI, 2 × SMS rs-EPI, and 3 × SMS rs-EPI (*p* < 0.05), while there were no significant different between 2 × SMS rs-EPI and rs-EPI (all *p* > 0.05). In the b = 1000 s/mm^2^ DWI image of the 3 × SMS rs-EPI sequence, parallel acquisition artifacts can be seen, and rectal structures were poorly visualized (Table 3 and Figure 2andFigure 3).

### 3.3. Quantitative Image Quality Analysis

The SNR and CNR on DWI images were significantly different among three rs-EPI sequences (all *p* < 0.001), while these were not significantly different between 2 × SMS rs-EPI and rs-EPI (all *p* > 0.001). Geometric distortions on DWI images were statistically different among the three groups of sequences and the geometric distortion of 2 × SMS rs-EPI was the lowest (Table 4).

The inter-observer agreement was excellent for ADC_lesion_ (ICC = 0.931 (0.895, 0.955), 0.963 (0.942, 0.976), and 0.854 (0.775, 0.906)) for rs-EPI, 2 × SMS rs-EPI, and 3 × SMS rs-EPI, respectively. The inter-observer agreement was excellent for ADC_normalized_ (ICC = 0.889 (0.825, 0.930), 0.885 (0.760, 0.938), and 0.854 (0.775, 0.906)) for rs-EPI, 2 × SMS rs-EPI, and 3 × SMS rs-EPI, respectively.

In the comparison between rectal tumor tissues of ADC values, there was a significant difference among three rs-EPI sequences (1.19 ± 0.21 vs.1.18 ± 0.22 vs.1.10 ± 0.26, *p* < 0.001). In the comparison between rectal normal tissues of ADC values, there was a significant difference among three rs-EPI sequences (1.68 ± 0.13 vs. 1.66 ± 0.14 vs. 1.54 ± 0.20, *p* < 0.001). For rectal tumor tissue and normal rectal tissue, there were no significant differences between the 2 × SMS rs-EPI and rs-EPI sequences (*p* = 0.068/*p* = 0.066); There was a significant difference of ADC values between 3 × SMS rs-EPI and 2 × SMS rs-EPI, 3 × SMS rs-EPI and rs-EPI (all *p* < 0.001). With the increase of the acceleration factor, the ADC values of both normal rectal tissue and tumor tissue decreased, as shown in Table 5 and Figure 4.

## 4. Discussion

Our study found that, using an AF of 2, 2 × SMS rs-EPI allowed for a considerable reduction of acquisition time while maintaining high diagnostic image quality and lesion conspicuity. When increasing the AF to 3, a significant deterioration in image quality was observed in patients with reduced visibility of all organ parts in the diffusion weighted images and reduced image quality of the corresponding ADC maps. Moreover, artefacts and signal inhomogeneity in 3 × SMS rs-EPI were significantly increased, which led to a poorer overall image quality and to a limited lesion conspicuity, thus discarding the use of higher AFs for clinical routine applications in its current implementation. Based on these results, further evaluation of AFs higher than 3 was waived.

The scan acquisition time can be accelerated by increasing the acceleration factor of SMS, while an excessive acceleration factor is expected to be accompanied by a deterioration in image quality. Therefore, the most important thing is to find the balance between shortening the scanning time by SMS AF_S_ and obtaining the best image quality. In our study, we found that 2 × SMS rs-EPI can reduce the scanning time while obtaining better image quality, since there was no difference in scores of overall image quality, lesion conspicuity, and margin sharpness of the DWI images and ADC map compared with rs-EPI. These results are the same as those previously reported [11,18,19]. Neither the parotid gland tumors nor nasopharyngeal carcinoma were affected by the significant motion of the organ, and a better image quality can be obtained using 2 × SMS rs-EPI, but the rectum is more affected by intestinal peristalsis; 2 × SMS rs-EPI can still obtain stable, high-quality images. However, a significantly worse overall image quality was observed, for lesion conspicuity and margin sharpness, between 3 × SMS rs-EPI and the other two sequences. SMS acceleration includes acceleration factor phase encoding (PE) and SMS factor. Excessive SMS acceleration factor reduced the intra-slice SNR, making it impossible for the GRAPPA algorithm used in acceleration factor PE to decode the parallel acquired data and create artifacts. Therefore, although an acceleration factor of 3 can significantly reduce the scan time, it is not feasible in rectal applications.

Tu, C et al. [18] compared the CNR and lesion distortion between rs-EPI, 2 × SMS rs-EPI and 3 × SMS rs-EPI in 105 nasopharyngeal carcinomas and reported that no significant differences were found in the CNR and lesion distortion between rs-EPI and 2 × SMS rs-EPI; however, the 3 × SMS rs-EPI shows the lowest CNR and lesion distortion. In this study, we found similar CNR but less geometric distortion in 2 × SMS rs-EPI vs. rs-EPI, similar to Jiang et al.’s study [19]. It is important to reduce distortion in rectal cancer MR imaging, and a true rectal display helps us to observe the circumferential resection margin (CRM) status, epidural vascular invasion (EMVI), depth of epidural invasion (EVI), discontinuous epidural vascular spread/deposition, and mucus presence; besides the T and N stages, these high-risk features also facilitate clinical decision making and can be used to classify rectal cancer patients according to their prognosis [23,24]. Our study result indicated that the SMS technique with an acceleration factor of 2 reduces the scan time without compromising image quality and with less distortion in the rectum, which can increase the clinical usability of rs-EPI for assessing rectal cancers.

The ADC values of rectal cancer may reflect cancer aggressiveness, which can predict the histologic T stage, differentiation, and therapeutic response after chemoradiotherapy [25]. Several studies reported no significant differences in ADC values between 2 × SMS ss-EPI and conventional ss-EPI in several organs [26,27,28], including the rectum [11]. Our study also revealed that there was no significant difference of ADC values between rs-EPI and 2 × SMS rs-EPI in the tumor tissue and normal tissue of the rectal cancer. A few previous studies have reported decreased ADC values using SMS-DWI in the liver and pancreas in comparison with conventional DWI [15,22,29,30]. These studies hypothesized that the elevated noise floor in SMS-DWI may result in generally lower ADC values [22]. However, several subsequent studies reported no significant differences in ADC values between SMS-accelerated sequences and conventional DWI in several organs [26,27,28], which is in line with our results. This difference may be explained by the fact that the TR was chosen to be as low as possible to minimize SMS technology, resulting in a lower SNR of DWI images and unstable calculated ADC values [22,30].

Our study had limitations. First, this was a retrospective, single-center study with a small sample size. Second, the study population included only patients with biopsy-confirmed rectal cancer, while further studies should include patients with benign rectal lesions or those who received radiotherapy for rectal cancer to validate the study findings further. Third, manual human measurement of ROI values increases the likelihood of sample error. Fourth, our study only examines the acceleration factor and does not discuss the impact of slice distance and number of shots on image quality, it will be designed and added prospectively in subsequent studies.

## 5. Conclusions

SMS rs-EPI using an AF of 2 is feasible for rectal MRI resulting in substantial reductions in acquisition time while maintaining diagnostic image quality and similar ADC values to those of rs-EPI when the slice distance and number of shots are the same among three rs-EPI sequences.

## Figures and Tables

**Figure 1 diagnostics-13-00474-f001:**
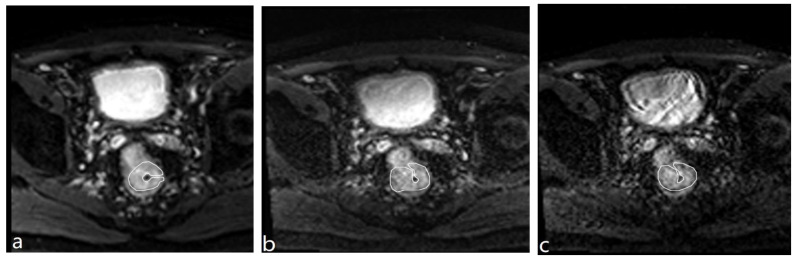
The ROI drawing for cancer lesions on the b = 0 s/mm^2^ maps of rs-EPI (**a**); 2 × SMS rs-EPI (**b**), 3 × SMS rs-EPI (**c**).

**Figure 2 diagnostics-13-00474-f002:**
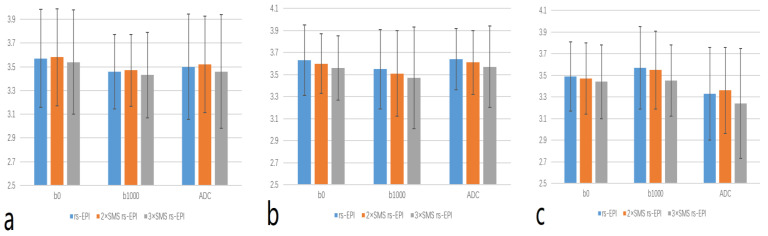
Comparison of qualitative evaluation metrics of the three EPI sequences. (**a**) lesion conspicuity; (**b**) overall image quality; (**c**) margin sharpness.

**Figure 3 diagnostics-13-00474-f003:**
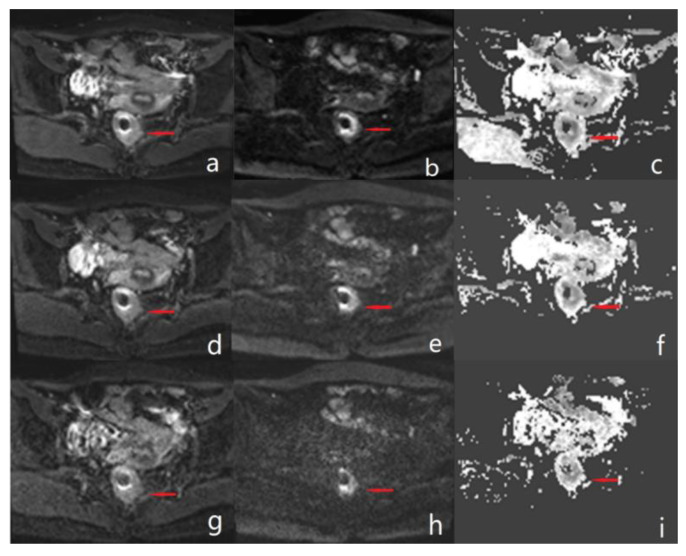
Male, 53 years old with rectal cancer. (**a**–**c**) Diffusion-weighted and ADC maps of conventional rs-EPI sequences with b-values of 0 s/mm^2^ and 1000 s/mm^2^, respectively. (**d**–**f**) Diffusion-weighted and ADC maps of 2 × SMS rs-EPI sequences with b-values of 0 s/mm^2^ and 1000 s/mm^2^, respectively. (**g**–**i**) Diffusion-weighted and ADC maps of 3 × SMS rs-EPI sequences with b-values of 0 s/mm^2^ and 1000 s/mm^2^, respectively. The arrow in the figure points to a rectal cancer lesion.

**Figure 4 diagnostics-13-00474-f004:**
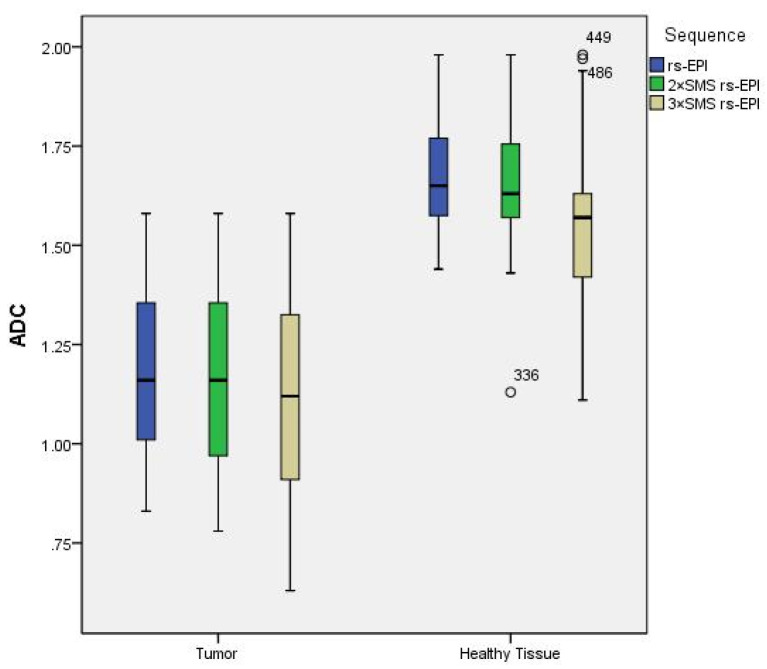
Box plot of the differences between ADC values (10^−3^ mm^2^/s) in tumors and healthy tissue measured using the three sequences.

**Table 1 diagnostics-13-00474-t001:** The parameters of three DWI sequences.

Parameters	rs-EPI	2 × SMS rs-EPI	3 × SMS rs-EPI
b value (s/mm^2^)	0.1000	0.1000	0.1000
Number of Readout segments	5	5	5
Fat press method	SPAIR	SPAIR	SPAIR
TR (ms)	5000	2270	1800
TE (ms)	51	52	54
Reversal time (s)	210	210	210
FOV (mm)	216 × 216	216 × 216	216 × 216
Matrix	128 × 128	128 × 128	128 × 128
Voxel size (mm^3^)	1.7 × 1.7 × 4.5	1.7 × 1.7 × 4.5	1.7 × 1.7 × 4.5
thickness (mm)	4.5	4.5	4.5
Distance Factor (mm)	0.45	0.45	0.45
Number of slices	24	24	24
iPAT	GRAPPA 2	GRAPPA 2	GRAPPA 2
Acceleration factor		2	3
Acquisition time (s)	137	60	51

**Table 2 diagnostics-13-00474-t002:** Clinicopathological characteristics of the patients.

Clinicopathological Characteristics		*n* Percentile (%)
Gender	Male	55 (66.27)
	Female	28 (33.73)
Age (years)		58.82 (11.71)
BMI (kg·m^−2^)		22.82 (1.91)
Location of tumor	Upper-middle segment	34 (40.96)
	Lower segment	49 (59.04)
differentiation grade	well	13 (15.7)
	moderate	64 (77.1)
	poor	6 (7.2)
CEA	+	55 (66.27)
	−	28 (33.73)
CA19-9	+	40 (48.19)
	−	43 (51.81)
Nerve invasion	+	56 (67.5)
	−	27 (32.5)
Vascular cancer embolus	+	61 (73.5)
	−	22 (26.5)
Cancer nodules	+	68 (81.9)
	−	15 (18.1)

**Table 3 diagnostics-13-00474-t003:** Comparison of three rs-EPI sequences showing subjective scores of rectal image quality.

Qualitative Parameters	rs-EPI	2 × SMS rs-EPI	3 × SMS rs-EPI	*X*^2^ Value	*p*-Value
Lesion Conspicuity					
b = 0 s/mm^2^	3.57 ± 0.41	3.58 ± 0.41	3.54 ± 0.44 ^ab^	11.143	0.004
b = 1000 s/mm^2^	3.46 ± 0.31	3.47 ± 0.30	3.43 ± 0.36 ^ab^	9.579	0.008
ADC map (×10^−3^ mm^2^/s)	3.50 ± 0.44	3.52 ± 0.41	3.46 ± 0.48 ^ab^	6.909	0.032
Overall image quality					
b = 0 s/mm^2^	3.63 ± 0.32	3.60 ± 0.27	3.56 ± 0.29 ^ab^	9.579	0.008
b = 1000 s/mm^2^	3.55 ± 0.36	3.51 ± 0.39	3.47 ± 0.46 ^ab^	11.273	0.004
ADC map (×10^−3^ mm^2^/s)	3.64 ± 0.28	3.61 ± 0.29	3.57 ± 0.37 ^ab^	11.273	0.004
Margin sharpness					
b = 0 s/mm^2^	3.49 ± 0.32	3.47 ± 0.33	3.44 ± 0.34	18.488	<0.001
b = 1000 s/mm^2^	3.57 ± 0.38	3.55 ± 0.36	3.45 ± 0.33 ^ab^	14.392	0.001
ADC map (×10^−3^ mm^2^/s)	3.33 ± 0.43	3.36 ± 0.40	3.24 ± 0.51 ^ab^	18.588	<0.001

Note: ^a^ compared with rs-EPI, *p* < 0.05; ^b^ compared with 2 × SMS rs-EPI, *p* < 0.05.

**Table 4 diagnostics-13-00474-t004:** Comparison of quantitative evaluation indexes of three EPI sequences.

	Rs-EPI	2 × SMS rs-EPI	3 × SMS rs-EPI	*X*^2^ Value	*p* Value
SNR					
b = 0 s/mm^2^	120.95 ± 21.66	119.52 ± 23.27	111.88 ± 19.31 ^ab^	36.105	<0.001
b = 1000 s/mm^2^	53.31 ± 7.15	52.79 ± 7.58	52.42 ± 8.09 ^ab^	13.040	0.001
CNR					
b = 0 s/mm^2^	4.84 ± 0.47	4.88 ± 0.47	4.74 ± 0.55 ^ab^	32.109	<0.001
b = 1000 s/mm^2^	4.26 ± 0.67	4.31 ± 0.67	4.18 ± 0.75 ^ab^	18.588	<0.001
Geometric distortion					
b = 0 s/mm^2^	3.49 ± 0.32	3.47 ± 0.32	3.51 ± 0.33 ^ab^	13.231	0.001
b = 1000 s/mm^2^	3.57 ± 0.38	3.52 ± 0.33	3.61 ± 0.43 ^ab^	25.529	<0.001

Note: ^a^ compared with rs-EPI, *p* < 0.05; ^b^ compared with 2 × SMS rs-EPI, *p* < 0.05.

**Table 5 diagnostics-13-00474-t005:** The comparison of ADC value between three EPI sequences.

	rs-EPI	2 × SMS rs-EPI	3 × SMS rs-EPI	*X*^2^ Value	*p* Value
ADC_Lesion_ (10^−3^ mm^2^/s)	1.19 ± 0.21	1.18 ± 0.22	1.10 ± 0.26 ^ab^	40.582	<0.001
ADC_Normalized_ (10^−3^ mm^2^/s)	1.68 ± 0.13	1.66 ± 0.14	1.54 ± 0.20 ^ab^	51.835	<0.001

Note: ^a^ compared with rs-EPI, *p* < 0.05; ^b^ compared with 2 × SMS rs-EPI, *p* < 0.05.

## Data Availability

The data is unavailable due to privacy or ethical restrictions.

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
