# Peer review of "Feasibility of Simultaneous Multislice Acceleration Technique in Readout-Segmented Echo-Planar Diffusion-Weighted Imaging for Assessing Rectal Cancer"

_diagnostics, 2023, doi:10.3390/diagnostics13030474_

Round 1
Reviewer 1 Report (New Reviewer)
This paper aims to find an optimal acceleration between readout segmented and simultaneous multi-slice (SMS) acquisitions for clinical diffusion weighted imaging applications. However, the parameter optimization is not validated on generalization, and some parts need to be clarified.
1. The SMS imaging combined with multi-shot segmented acquisition along the phase encoding direction has been more commonly used rather than with readout segmented. It is not clear why you didn’t use multi-shot segmented approach as a reference. Additionally, the comparison between different segmented approaches (readout and phase encoding) should be described in this work.
2. The number of readout-segmentation is very susceptible to the patient motions resulting from shot-to-shot inconsistency. Please describe the above issues of motion and inconsistencies during readout acquisition in more details in the discussion section.
3. The reconstruction quality of SMS imaging is very sensitive to the distance between the multiple slices. This work needs to be validated on varying the distance between slices in terms of parameter optimizations. Additionally, the manuscript shows the distance factor, but will be more clear if the authors use mm unit as a distance.
4. Parallel imaging and SMS imaging compete with one another. This work shows the comparison images with increasing slice acceleration with a fixed inplane acceleration. However, to find an optimized acceleration, the authors need to add the comparison with increasing inplane acceleration while keeping SMS factor fixed. That is, this work needs to be validated on the bi-directional acceleration between in-plane and through-plane at the same time.
5. This work seems to show abdomen for clinical applications. The acquisition mode is free-breathing not breadth-hold. However, the abdomen is one of the areas where motion artifacts is very severe even with breath-hold imaging. Is there any reason to choose breath-hold mode during acquisition? Additionally, how did you address the motion issue with free-breathing mode.
To conclude, this work needs further analysis before it can be recommended for publication.
Author Response
Please see the attachment

Reviewer 2 Report (New Reviewer)
The abstract needs quantification. The Introduction needs more explanation. SNR, CNR analysis and Statistical analysis need a relook and more information. The results and discussion part has to be enhanced. Some more statistical tests may be included. The authors presented the cause and effect but not the universal ideas which is generalization. Conclusion will be added. Even though the authors had done a pain staking work and the presentation needs improvement.
Author Response
Reviewer 2:
The abstract needs quantification. The Introduction needs more explanation. SNR, CNR analysis and Statistical analysis need a relook and more information. The results and discussion part has to be enhanced. Some more statistical tests may be included. The authors presented the cause and effect but not the universal ideas which is generalization. Conclusion will be added. Even though the authors had done a pain staking work and the presentation needs improvement.
Response: Thanks for your advice. We have done our best to revise the manuscript. I have rewritten the abstract, intro, result, and discussion. The previous description did have a lot of unclear points, thanks to the reviewers for reminding us. I have revised the manuscript in word revision mode, please review it. The inter-observer agreement test between the two observers replaces the Kappa test with the ICC. For the other statistical methods, we referred to similar previously published literature[1-3] that used the same methods as we did.
- Park, J.H.; Seo, N.; Lim, J.S.; Hahm, J.; Kim, M.J. Feasibility of Simultaneous Multislice Acceleration Technique in Diffusion-Weighted Magnetic Resonance Imaging of the Rectum. Korean J Radiol 2020, 21, 77-87, doi:10.3348/kjr.2019.0406.
- Tu, C.; Shen, H.; Liu, D.; Chen, Q.; Yuan, X.; Li, X.; Wang, X.; Liu, R.; Wang, X.; Li, Q.; et al. Simultaneous multi-slice readout-segmentation of long variable echo-trains for accelerated diffusion-weighted imaging of nasopharyngeal carcinoma: A feasibility and optimization study. Clin Imaging 2021, 79, 119-124, doi:10.1016/j.clinimag.2021.04.009.
- Taron, J.; Martirosian, P.; Kuestner, T.; Schwenzer, N.F.; Othman, A.; Weiss, J.; Notohamiprodjo, M.; Nikolaou, K.; Schraml, C. Scan time reduction in diffusion-weighted imaging of the pancreas using a simultaneous multislice technique with different acceleration factors: How fast can we go? Eur Radiol 2018, 28, 1504-1511, doi:10.1007/s00330-017-5132-1.
Round 2
Reviewer 1 Report (New Reviewer)
I am disappointed that authors did not perform additional validation in other parameters where such technique will be more useful. It clearly has many limitations in claiming the generalization. There are already several parameters available even with Siemens scanner for the slice distance and number of shots, and etc. They should at least recognize the limitations more clearly and tone down the significance. But, any issue not addressed in the response.
Author Response
Thanks for your advice. We also agree that this is a meaningful aspect, but the current data set can no longer add slice distance and number of shots parameters to the comparison on the same subjects, we recognize this as a limitation and have written it into the article limitation, which will be prospectively designed and added again in the subsequent study. And our conclusion also holds only if the slice distances and number of shots of all three sequences are the same.

Reviewer 2 Report (New Reviewer)
All the corrections are included in the paper. Hence, the paper may be accepted. However, the author may write a separate conclusion.
Author Response
Thanks for your attention, the separate conclusion has been added to the manuscript.
This manuscript is a resubmission of an earlier submission. The following is a list of the peer review reports and author responses from that submission.
Round 1
Reviewer 1 Report
The study entitled "Feasibility of Simultaneous Multislice Acceleration Technique in readout-segmented echo-planar imaging for assessing rectal cancer" enrolled 83 patients with rectal cancer and compared the lesion conspicuity, overall image quality, sharpness, signal-to-noise ratio (SNR), and contrast-to-noise ratio (CNR) between conventional readout-segmented echo-planar imaging (rs-EPI), simultaneous multi-slice (SMS) rs-EPIs at 2 and 3 times acceleration. The results showed that the 2x accelerated SMS rs-EPI had equivalent lesion conspicuity, overall image quality, sharpness, SNR, and CNR with rs-EPI, and had shorter scan time than that of rs-EPI. In general, this study is similar to a recent paper that compared the EPI, 2x SMS-EPI, and 3x SMS-EPI in rectal cancers, but the only difference of this study was a rs-EPI used in this study. Both studies had similar conclusions that the 2x SMS with EPI and rs-EPI performed superior to the 3x SMS with EPI and rs-EPI, respectively. However, without a comparsion between 2x SMS EPI and 2x SMS rs-EPI, it is difficult for readers to know whether the 2x SMS rs-EPI is superior to 2x SMS EPI. Therefore, It is required to compare the differences between SMS EPI and SMS rs-EPI, such that the importance and novelty can be enhanced in this study.
1. Title should add "diffusion-weighted".
Abstract
2. In the purpose, "the optimal acceleration of SMS technoloy is also not determined in rectal cancer". In my opinion, the reference [16] has performed the comparison between 2x SMS EPI and 3x SMS EPI.
Introduction
3. In the line 1 of the fourth paragraph, "RS-EPI" should be corrected to "Rs-EPI".
Materials and Methods
4. In second paragraph, "the exclusion criteria were ... (3) serious image artifacts....". Because those image with serious image artifact may reflect the rs-EPI, 2x rs-EPI, and 3x rs-EPI, the comparison of images with artifacts may better demonstrate the feasibility of SMS rs-EPI in rectal cancer.
5. A comparison between SMS EPI and SMS rs-EPI may be needed to enhance the importance and novelty of this study.
MRI Imaging
6. "Conventional MRI flat scan sequences were..." What is "flat" scan?
7. "Tra" should be "transverse".
8. The b-values of 0 and 1000 s/mm2 were not suitable for rectal cancer. An intermediate b-value, such as 300-500 s/mm2, was more suitable to obtain an accurate ADC value in rectal cancer. The reference [16] utilized b-values of 0, 300, and 1000 s/mm2 to acquire DWI data.
Table 1
9. It is not clear whether parallel imaging was used for the three sequences in this study. If not, why?
10. "chemical element" should be corrected.
11. "SPAIR" should be spelled out.
12. How many segments were used for rs-EPI?
Image quality-qualitative analysis
13. "image storage and transfer system (PACS)" should be corrected.
Image quality-quantitative analysis
14. An example of ROI drawing for cancer lesions is helpful to demonstrate the quantitative analysis.
15. In the second paragraph, the equation for distortion rate should be corrected to (rs-EPI distance – T2WI distance)/T2WI distance*100.
Statistical analysis
16. ANOVA analysis with Bonferroni correction is needed to reduce Type I error in statistics.
Results
17. The number of male and female subjects in Table 2 were not consistent with the statements in the first paragraph.
18. Table 3 is not cited in the maintext.
19. Figure 1 is not cited in the results.
20. In Figure 1, the standard deviation should be added.
21. In Figure 2, the figure legends for (g-i) should be corrected.
22. In Figure 3, the unit for ADC is needed.
Discussion
23. In the second paragraph, "shorting" should be corrected to "shortening".
24. In the third paragraph, "PE" should be defined.
25. If GRAPPA cannot be used in SMS rs-EPI, the scan time is longer in SMS rs-EPI than SMS ss-EPI. When GRAPPA=2 was used, the distortion of SMS ss-EPI could be reduced and might be similar to 2x SMS rs-EPI without GRAPPA. However, it is not clear whether the 2x SMS rs-EPI was more suitable than the 2x SMS ss-EPI in reference [16] for the diagnosis of rectal cancers. Therefore, a comparison between 2x SMS ss-EPI and 2x SMS rs-EPI is needed.
26. In the fourth paragraph, "Chunrong" is not author's family name and should be corrected.
Conclusion
27. If there is no comparison between SMS ss-EPI and SMS rs-EPI, the conclusion of "using SMS rs-EPI sequences may further promote a more accurate diagnosis of rectal cancer" cannot be made based on the present study.
